# Adherence to intravenous chemotherapy and associated factors among patients with cancer at Hawassa University Comprehensive Specialized Hospital Cancer Treatment Center, Sidama Region, Southern Ethiopia

**Bargude Balta⊙*, Alemayehu Delelegn, Gulema Demissie, Bedilu Deribe**

Hawassa University College of Medicine and Health Sciences, Hawassa, Ethiopia

* barjuda@gmail.com

## Abstract

### Background

Medication adherence refers to how closely a patient follows the prescribed timing and dosage of their treatment. Adherence to chemotherapy is particularly complex and multi-faceted, and it can have a significant impact on the effectiveness of the therapy. In Ethiopia, non-adherence to chemotherapy is on the rise, but there has been limited research specifically on adherence to intravenous (IV) chemotherapy.

### Objective

This study assesses IV chemotherapy adherence and associated factors among cancer patients in the Sidama Region, southern Ethiopia.

### Methods

A hospital-based cross-sectional study included a purposive sample of 413 cancer patients undergoing IV chemotherapy. The Morisky Medication Adherence Scale was used to measure adherence levels. Data analysis was performed using SPSS version 26, employing descriptive statistics such as frequency distribution, mean, median, and standard deviation to describe the characteristics and magnitude of IV chemotherapy adherence. Bivariable and multivariable logistical regression analysis was conducted to identify factors associated with IV chemotherapy adherence.

### Result

The current study revealed that the overall magnitude of good adherence toward chemotherapy treatment among cancer patients was 176/413(42.6%), with a 95% CI of 38–47.9. The multivariable analysis identified several independent factors associated

**Data availability statement:** Data cannot be shared publicly because of ethical restrictions due to data containing potentially identifying or sensitive patient information (Even de-identified). Data are available from the authors after appropriate request to the School of Nursing and Midwifery Research Ethics Committee, Hawassa University contact: (oerc@hu.edu.et) for researchers who meet the criteria for access to confidential data.

**Funding:** The author(s) received no specific funding for this work.

**Competing interests:** The authors have declared that no competing interests exist.

**Abbreviations:** AC, Adriamycin And Cyclophosphamide; CTC, Chemotherapy Treatment Center; CAPOX, Capecitabine And Oxaliplatin; ECOG, Eastern Cooperative Oncology Group; FOLFOX, Folic Acid, Fluorouracil and Oxaliplatin; GLOBOCAN, Global Cancer Observatory; IV, Intravenous; ASIR, Age-Specific Incidence Rate; MMAS, Morisky Medication Adherence Scale

with IV chemotherapy adherence. These factors included being married (AOR = 3.2, 95% CI:1.3,8), employment as a government employee (AOR = 2.4,95% CI:1.3,4.5), availability of transportation (AOR = 5.5, 95% CI:2.1,14), Good social support (AOR = 2.1,95% CI:1.9,11.3) and having a curative goal of treatment (AOR = 1.7,95% CI:1.1,2.7).

## Conclusion

In the current study, the magnitude of intravenous chemotherapy adherence among patients with cancer was low compared to published national and international findings. Possible contributing factors include marital status, employment, Transportation availability, and social support. Targeted measures are needed to improve adherence and, as a result, maximize therapeutic benefits.

## Background

Adherence to intravenous (IV) chemotherapy is defined as the extent to which a patient's behavior corresponds to the intravenous chemotherapy prescribed by their doctor, and it is typically measured as the percentage of doses received relative to those suggested. The most common limit for defining adherence in these terms is 80% [1]. It is also known as compliance or "the degree to which a patient complies with medical recommendations" [2].

The magnitude of chemotherapy adherence varies widely, with studies estimating adherence rates between 50% and 80%, depending on factors such as cancer type, treatment regimen, and patient characteristics [3]. In Ethiopia, chemotherapy adherence was 42.3% which varies widely depending on the type of cancer and patient characteristics [4]. Factors influencing adherence include sociodemographic determinants such as age, gender, and education level, as well as clinical factors like the presence of comorbidities, severity of side effects, and availability of medications [4,5].

Chemotherapy adherence is essential for successful cancer treatment, as it directly impacts the effectiveness of therapy and the patient's prognosis. However, several factors can hinder adherence, including the severity of side effects, emotional distress, financial challenges, and lack of social or psychological support. Physical side effects, such as nausea, fatigue, and pain, may discourage patients from completing their treatment as prescribed. Financial constraints, such as the cost of medications and treatment-related expenses, may further complicate adherence, especially in underinsured or low-income patients [6]. Effective interventions, such as patient education, adherence monitoring, and tailored support programs, have shown promise in improving adherence rates [7].

According to US-based report Medication, non-adherence accounts for between 33 and 67% of medication-related hospitalizations, costing the economy $100 billion a year. About 50% of individuals with chronic illnesses do not take their prescription drugs as directed [8]. An African-based study revealed that the high frequency of poor adherence to cytotoxic drugs was mainly associated with low income levels [9].

Research findings show that individuals who received less than 85% of the recommended amount of intravenous chemotherapy had a worse prognosis than those who received the full course of treatment [10]. Chemotherapy adherence is a major determinant of treatment efficacy and optimal clinical benefit. Poor chemotherapy adherence leads to drug resistance and recurrence [11].

Adherence may result in lower medical expenses, fewer crises, relapses, or exacerbations of disease, improved patient quality, and life preservation [12]. Adherence to chemotherapy is

mainly related to the patient's understanding of the treatment and ability to remember information provided by the physician, treatment length and psychological distress, treatment cost [13].

In Ethiopia, where access to chemotherapy is very limited, there is also no well-documented evidence of adherence to chemotherapy among adult cancer patients. The chemotherapy adherence challenge is becoming a major public health concern, especially in Ethiopia, its magnitude is increasing but little has been done about adherence to chemotherapy treatment [4].

Furthermore, a clearer articulation of the research gap is essential to identify unmet needs in chemotherapy adherence studies and inform future interventions. Thus, this study aimed to determine adherence to cancer chemotherapy and the associated factors among patients with cancer at HUCSH Sidama Region Southern Ethiopia. These findings will inform the interventions aimed at improving adherence to care among cancer patients in Ethiopia.

## Methods

### Study area and period

This study was conducted at Hawassa University Comprehensive Specialized Hospital Cancer Treatment Center (HUCSH-CTC), which is located in Hawassa City. Hawassa City serves as the capital city of Sidama Regional State. It is located 273 km south of Addis Ababa. According to the Hawassa City Administration Health Department in the 2013 population profile, the total population of Hawassa city is 394,057. HUCSH-CTC is located in Hawassa; the capital town of Sidama Regional State HUCSH is a teaching, comprehensive tertiary hospital with approximately 520 inpatient beds.

HUCSH is the largest and most well-known public hospital in the region. The hospital hosts a Cancer Treatment Center out of its main hospital. The Center is housed in a separate, newer building on hospital grounds. It consists of three floors: the clinic with three examination rooms, physicians' offices, waiting room, radiation room, CT-scan room, pharmacy room, and laboratory unit; a floor for the inpatient ward; and a floor for meeting rooms, nurses' duty, and palliative care unit.

The center has Breast colorectal, lymphoma, gastric, and other gynecologic malignancies commonly diagnosed in HUCSH-CTC [2]. The center has sufficient beds, but scarcity of drugs and other medical instruments is highly predictive of treatment dropout. The study was conducted from May 20 to August 31, 2024.

### Study design

An institutional-based cross-sectional study was employed.

### Population

**Source population.** All adults with cancer on medical follow-up at HUCSH-CTC, and were willing to participate were included.

**Study population.** All adults ≥ 18 years diagnosed with cancer and treated with IV chemotherapy at HUCSH-CTC

### Study subjects

All adult patients ≥ 18 years of age who took at least one cycle of chemotherapy at HUCSH_CT during the study period.

### Eligibility criteria

**Inclusion criteria.** All cancer patients who received at least one cycle of IV chemotherapy treatment were recorded in the chart.

**Exclusion criteria.** All cancer patients who were severely ill, and unable to communicate were excluded from the study.

## Sample size determination

Single population proportion formula was used to calculate the sample size with a 95% confidence interval (α is 0.05 with 95% CI, Zα/2 = 1.96), the margin of error (d) 5%, and an estimate of proportion (p) of IV adherence study done in the Amhara Region, Northeastern Ethiopia was 42.3% [4]. The formula was used as follows:

$$\text{n} = \frac{(Z-/2)^2 \, P(1-P)}{d^2} = \frac{(1.96)^2 \, 0.423(1-0.423)}{0.05^2} = 375$$

*Where:* - n = minimum sample size required for the study
Z= standard normal distribution (Z=1.96) with a confidence interval of 95%
P = proportion of adherence among cancer patients
d=is a tolerable margin of error (d=0.05)

The calculated sample size was 375, and after adding a 10 percent non-response rate, the final sample size was 413.

## Sampling procedure

We included all patients receiving chemotherapy during the study period purposely since all patients seen study period were 415.

## Study variables

**Independent variables.** **Socio-demographic:** age, sex, income, educational status, religion, residence, occupation, social support, health insurance,

**Cancer type-related factors:** Type of cancer, comorbidity, stage of cancer, family history of cancer, ECOG.

**Treatment-related factors:** distance treatment center, perceived fear of side effects, financial toxicity, presence of diarrhea, presence of pain, and root of chemotherapy.

**Social support and personal-related:** Social support, distance treatment center, health insurance, smoking, drinking alcohol, exercise, time to the hospital, transportation access, availability of drugs.

**Dependent variable.** Adherence to chemotherapy

**Operational definitions.** **Adherence to IV chemotherapy:** Based on the Morisky Medication Adherence Scale (MMAS-8) aggregate score of eight indicated good adherence and poor adherence when MMAS-8 with a sum score of 1–7. MMAS-8 self-report scale that evaluates medication-taking behavior established from the previously validated 4-item and augmented with extra questions to more accurately reflect obstacles related to compliance behavior [4].

**Social support:** is the ability of a patient's social network—which includes friends, neighbors, coworkers, family, and even pen pals and online relationships—to be perceived and received to lessen emotional strain, ease mental strain, and improve social adaptability. Mean scores on the total scale were calculated by summing items and dividing by 12. Higher mean scores indicate greater perceived social support and below mean indicate poor social support. The MSPSS has demonstrated good reliability and validity across various [4].

**Comorbidity:** According to the International Classification of Disease-10, the Charles Comorbidity Index was used during data collection.

## Data collection tool

Data were collected through a structured questionnaire developed from other studies to assess socio-demographic, personal, treatment-related, and cancer-related variables related to adherence to IV chemotherapy. The first section, such as socio-demographic characteristics (age, sex, income, educational status, religion, residence, occupation, social support, health insurance, and family history of cancer), was obtained directly from the patient via face-to-face interviews. The second part includes personal-related characteristics like drinking alcohol, smoking cigarettes, moderate exercise, traveling time to the hospital, distance to a treatment center, health insurance, distance to a treatment center, health insurance) which was collected from patient interviews the third part includes Cancer type-related factors including type of cancer, comorbidity, ECOG, types of cancer, and stage of cancer were obtained from medical charts at the cancer center.

The fourth section also deals with treatment-related factors like distance treatment centers, the goal of treatments, side effects, the presence of diarrhea, the presence of pain, and the root of chemotherapy which was retrieved from both patient and charts. Eight-item Morisky Medication Adherence Scale used to measure chemotherapy adherence [14] and twelve-item multidimensional scale used to measure perceived social support [15] which was collected by patient interviews.

## Data collection procedure

The data was collected through both face-to-face interviews and chart reviews using the Kobo toolbox, following the training of data collectors. Two data collectors with a BSc in nursing and prior experience in data collection were recruited, with one oncology nurse (MSc) supervising the collection. Training for data collectors and supervisors was provided by the principal investigator about the main principles that have to be respected during data collection and how each question in the questionnaires has to be understood. The overall data collection was supervised by the principal investigator

## Data quality assurance

Before data collection, a one-day orientation session was conducted for data collectors, covering data collection techniques, the purpose of data collection, and the content of the questionnaire. This session also guided how to approach respondents and handle potential difficulties. A pretest was done at Yanet Hospital, involving 5%of the total sample size, 2 weeks before the actual data collection. This pretest aimed to assess the instrument's simplicity, flow, and consistency, as well as to improve its reliability Each day, the filled questionnaires were checked for completeness by the supervisor. Before data entry and analysis, the data were rechecked for missing values by the principal investigator.

## Data processing and analysis

Data from eligible patients' medical records was extracted and imported into the Kobo toolbox. The data was then collected using Kobo and exported to SPSS version 26 for analysis. The level of chemotherapy adherence was calculated and assessed using descriptive statistics. The level of chemotherapy adherence was calculated and assessed using descriptive statistics. Before running the regression model, multi-collinearity diagnosis was conducted using variance inflation factors (VIF), with a value below 10 indicating no multicollinearity between two or more factors. Bivariable logistic regression analysis was conducted d to select variables for multivariable analysis. Variables with a $P$-value less than 0.05 were considered statistically significant. The adjusted odds ratio with its 95% confidence interval was used to demonstrate the strength of association between each explanatory variable and the outcome variable.

### Ethical and consent to participate

Ethical clearance (Ref: No. IRB/218/16) was obtained from the Research and Ethics Committee of the School of Nursing and Midwifery, College of Health Sciences, Hawassa University. Written informed consent was obtained from the study participants after explaining the objective of the study. Patient information was anonymized and kept confidential.

## Result

### Participant's Socio-demographic Characteristics

A total of 413 participants were interviewed, achieving a 100% response rate. The mean age of participants was 46.1 years, with a standard deviation of ± 14.5 years. The majority of study participants, 141 (34.1%), were 51 years or older. Over three-fourths of the participants, 322 (78%), were married. Most participants, 236 (57.1%), were urban inhabitants. Approximately 178 (43.1%) had no formal education. A total of 135 participants (32.7%) were farmers. Two-thirds of the participants, 269 (65.1%), had a monthly income of less than $35 (see Table 1).

**Table 1. Socio-demographic characteristics of cancer patients under IV chemotherapy treatment at HUCSH, Hawassa Sidama Regional State of Ethiopia, 2024 (n = 413).**

| Variables | Category | Frequency | (%) |
|---|---|---|---|
| Age of the respondent | 18-30 | 76 | 18.4 |
| | 31-40 | 95 | 23 |
| | 41-50 | 101 | 24.5 |
| | >51 | 141 | 34.1 |
| Sex | Male | 175 | 42.4 |
| | Female | 238 | 57.6 |
| Marital status | Married | 322 | 78 |
| | Single | 45 | 10.9 |
| | Divorced | 46 | 11.1 |
| Religion of the respondent | Orthodox | 167 | 40.4 |
| | Protestant | 127 | 30.8 |
| | Muslim | 111 | 26.9 |
| | Catholic | 8 | 1.9 |
| Residence | Urban | 236 | 57.1 |
| | Rural | 177 | 42.9 |
| Occupation of the respondent | Unemployed | 135 | 32.7 |
| | Government Employed | 82 | 19.9 |
| | Private employe | 73 | 17.7 |
| | Marchant | 69 | 16.7 |
| | Farmer | 54 | 13.1 |
| Educational status of the respondent | No formal education | 178 | 43.1 |
| | Primary | 109 | 26.4 |
| | Secondary | 70 | 16.9 |
| | Above Secondary | 56 | 13.6 |
| Monthly Income in USD | <35 | 269 | 65.1 |
| | ≥35 | 144 | 34.9 |

## Personal-related characteristics

The majority of participants, 358(86.7%), had no history of alcohol intake, and 375(90.8%) had no history of current smoking. Most of the participants, 400(96.9%), did not engage in moderate exercise. Almost half of the participants, 207 (50.1%), reported travel times of less than 3 hours. Nearly half, 227 (55%), reported travel distances of less than 100 kilometers (Table 2).

## Social support

More than half of the participants, 233 (56.4%), reported having good social support. A total of 245 participants (59.3%) strongly agreed that there is a special person who is around when they are in need during treatment, and 131 participants (31.7%) strongly agreed that there is a special person with whom they can share their joys and sorrows. Additionally, 256 participants (62%) reported that their family is willing to help them make decisions. More than half, 225 participants (54.5%), reported that there is a special person in their life who cares about their feelings, and 139 participants (33.7%) felt similarly (Table 3).

## Treatment-related characteristics

Two-thirds of study participants, 318 (77%), reported that chemotherapy was not available at treatment centers. Nearly one-third, 102 (24.7%), received Cyclophosphamide and doxorubicin treatment. The majority, 395 (95.6%), underwent chemotherapy alone, while most participants, 368 (89.1%), received polychemotherapy alone while most participants, 368 (89.1%), received polychemotherapy. Additionally, 176 participants (42.6%) reported undergoing four or more cycles of chemotherapy. Most of the cancer patients who participated in the study, 384 (93%), experienced side effects from their treatment (Table 4).

**Table 2. Personal-related characteristics of cancer patients under chemotherapy treatment at HUCSH, Hawassa Sidama Regional State of Ethiopia, 2024 (n = 413).**

| Variables | Category | Frequency | (%) |
|---|---|---|---|
| Drinking alcohol | Yes | 55 | 13.3 |
| | No | 358 | 86.7 |
| Smoking cigarette | Yes | 38 | 9.2 |
| | No | 375 | 90.8 |
| Moderate exercise | Yes | 13 | 3.1 |
| | No | 400 | 96.9 |
| Travel time to the hospital | < 3 Hrs. | 207 | 50.1 |
| | 3 to 6 hrs. | 89 | 21.5 |
| | >6 hrs. | 117 | 28.4 |
| Distance to treatment center | <100 Km | 227 | 55 |
| | 100-500 Km | 173 | 41.9 |
| | >500 Km | 13 | 3.1 |
| Health insurance | Yes | 205 | 49.6 |
| | No | 208 | 50.4 |
| Distance to treatment center | <100km | 227 | 55 |
| | 100-500 Km | 173 | 41.9 |
| | >500 Km | 13 | 3.1 |
| Health insurance | Yes | 180 | 49.6 |
| | No | 208 | 50.4 |

**Table 3. The overall social support level among cancer patients receiving chemotherapy at HUCSH, Sidama regional State of Ethiopia 2024(n = 413).**

| | Very strongly disagree | Strong disagree | Mildly disagree | Neu-tral | Very strongly agree | Strong agree | Mildly agree |
|---|---|---|---|---|---|---|---|
| | (%) | (%) | (%) | (%) | % | (%) | (%) |
| A special person who is around you when you need it. | 0.5 | 1.0 | 1.0 | 1.5 | 24.0 | 59.3 | 12.8 |
| Special person, you share joys and sorrows | 1.2 | 2.9 | 7.5 | 3.9 | 15.5 | 31.7 | 37.7 |
| My family tries to help me. | 5 | 1.0 | 1.2 | 0.5 | 30.3 | 53.3 | 13.3 |
| I get the emotional support from family. | 1.4 | 2.4 | 7.5 | 2.2 | 20.1 | 33.2 | 33.4 |
| A special person who is a source of comfort to me | 0.5 | 0.7 | 1.0 | 2.4 | 32.7 | 47.5 | 15.3 |
| My friends try to help me. | 3.4 | 4.8 | 16.5 | 4.1 | 6.5 | 25.7 | 39.0 |
| Count friends when things go wrong. | 10.2 | 17.7 | 10.2 | 12.1 | 2.9 | 13.3 | 33.7 |
| Talk about my problems with my family. | 0.5 | 2.4 | 15.0 | 1.9 | 16.9 | 38.7 | 24.5 |
| Friends with whom I can share my joys and sorrows. | 8.5 | 16.0 | 16.7 | 8.5 | 5.6 | 14.3 | 30.5 |
| The special person in my life who cares about my feelings. | 0.2 | 1.2 | 4.8 | 2.2 | 11.1 | 54.5 | 25.7 |
| My family is willing to help me make decisions. | | 0.5 | 0.7 | 0.5 | 27.1 | 62.0 | 9.2 |
| I can talk about my problems with my friends. | 5.8 | 9.2 | 18.2 | 4.6 | 2.7 | 10.9 | 48.7 |

**Table 4. Treatment-related characteristics of cancer patients under chemotherapy treatment at HUCSH, Hawassa, Sidama Regional State of Ethiopia, 2024 (n = 413).**

| Variable | Category | Frequency | Percent |
|---|---|---|---|
| Chemotherapy available | Yes | 95 | 23 |
| | No | 318 | 77 |
| Type of chemotherapy | Cyclophosphamide and Doxorubicin | 102 | 24.7 |
| | AC-Taxol | 62 | 15 |
| | FOLFOX | 72 | 17.4 |
| | Paclitaxel + cisplatin | 50 | 12.1 |
| | CAPOX | 55 | 13.3 |
| | Others | 72 | 17.4 |
| Treatment modalities | Chemotherapy only | 395 | 95.6 |
| | Chemotherapy and surgery | 12 | 2.9 |
| | Chemotherapy plus hormonal | 6 | 1.5 |
| Number of chemotherapies | Mon chemotherapy | 45 | 10.9 |
| | Polychemotherapy | 368 | 89.1 |
| Chemotherapy cycle | 1st cycle | 51 | 12.3 |
| | 2nd cycle | 93 | 22.5 |
| | 3rd cycle | 93 | 22.5 |
| | 4th and above cycle | 176 | 42.6 |
| Side effect | Yes | 384 | 93 |
| | No | 29 | 7 |
| Goal of treatment | Palliative | 272 | 65.9 |
| | Curative | 141 | 34.1 |

## Cancer-related characteristics

The majority of cancer patients, 330(79.9%). had no family history of cancer. Nearly half, 195(47.2%), were in stage four of the disease, and 102(24.7%) were diagnosed with breast cancer. Most participants, 331 (80.1%), had no history of comorbidity. One-fourth, 23

participants (28%), reported having multiple comorbidities. Almost half of the patients, 193(46.7%), had an ECOG (Eastern Cooperative Oncology Group) performance status of 0 (Table 5).

## Magnitude of Adherence Level

In this study, the magnitude of adherence among cancer patients receiving chemotherapy was 176/413(42.6%) with 95% (CI: 38–47.9) indicating good adherence, as shown in Fig 1.

## Bivariable and multivariable analysis of factors associated with chemotherapy adherence

Bivariable and multivariable regression models were used to identify the independent factors associated with chemotherapy adherence. The Bivariable regression model showed significant associations between chemotherapy adherence and factors such as marital status, occupation, insurance, transportation availability, pain, social support, and the goal of treatment. After multivariable analysis, the following factors were identified as independently associated with chemotherapy adherence: being married (AOR = 3.2, 95% CI:1.3,8), Gov't employee (AOR = 2.4,95% CI:1.3,4.5), Transportation availability (AOR = 5.5, 95% CI:2.1,14), Good social support (AOR = 2.1,95% CI:1.9,11.3) and curative goal of treatment (Table 6).

**Table 5. Treatment-related characteristics of cancer patients under chemotherapy treatment at HUCSH, Hawassa, Sidama Regional State of Ethiopia, 2024 (n = 413).**

|  | Category | Frequency | Percent |
|---|---|---|---|
| Family history of cancer | Yes | 83 | 20.1 |
|  | No | 330 | 79.9 |
| Type of cancer | Breast | 102 | 24.7 |
|  | Colorectal | 61 | 14.8 |
|  | Cervical | 33 | 8 |
|  | Lymphoma | 30 | 7.3 |
|  | Lung | 29 | 7 |
|  | Gastric | 27 | 6.5 |
|  | Esophageal | 25 | 6.1 |
|  | Others | 106 | 25.7 |
| Stage of cancer | Stage 1 | 93 | 22.5 |
|  | Stage 2 | 46 | 11.1 |
|  | Stage 3 | 79 | 19.1 |
|  | Stage 4 | 195 | 47.2 |
| Comorbidity | Yes | 82 | 19.9 |
|  | No | 331 | 80.1 |
| Type of comorbidity | DM | 17 | 20.7 |
|  | Hypertension | 13 | 15.9 |
|  | Multiple comorbidity | 23 | 28 |
|  | Respiratory problem | 15 | 18.3 |
|  | Other | 14 | 17.1 |
| ECOG | 0 | 193 | 46.7 |
|  | 1 | 173 | 41.9 |
|  | 2 | 27 | 6.5 |
|  | 3 | 20 | 4.8 |

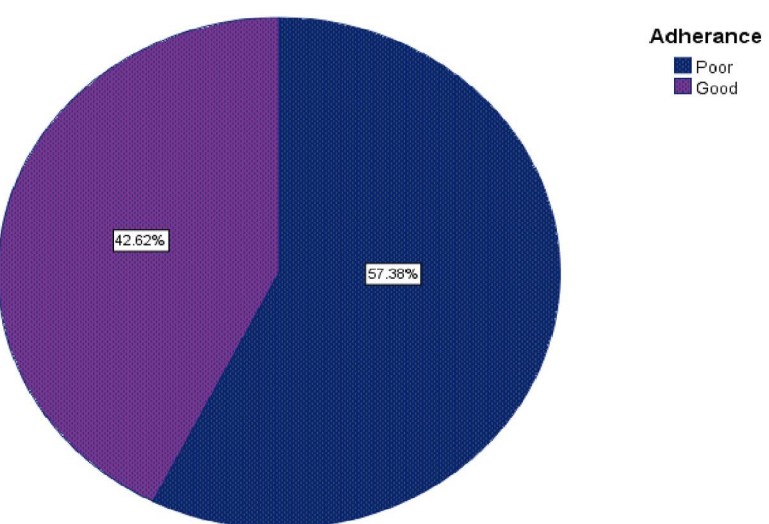

**Fig 1. Pie chart shows cancer patients' level of adherence among cancer patients receiving chemotherapy at HUCSH, Sidama Regional State 2024 (n =413).**

Table 6. Bivariable and multivariable analysis of factor-associated chemotherapy adherence among cancer patients at HUCSH, Hawassa Sidama Regional State of Ethiopia, 2024 (n=413).

| Variable | Adherence | | COR (95% C. I) | P-Value | AOR (95% C.I) | P-Value |
|---|---|---|---|---|---|---|
| | Poor | Good | | | | |
| Married | 177 | 145 | 2.9(1.2,7) | 0.004 | 3.2(1.3,8) | 0.01 |
| **Marital status** | 24 | 21 | 3.5(1.5-8.7) | 0.03 | 4.3(1.4,13.5) | 0.07 |
| Married | 36 | 10 | 1 | | 1 | |
| Single | | | | | | |
| Divorced | 81 | 54 | 1 | | 1 | |
| **Occupation** | 38 | 31 | 1.2(0.7,2) | 0.5 | 1.7(0.8,3) | 0.14 |
| Unemployed | 29 | 53 | 2.7(1.5.4.8) | 0.001 | 2.4(1.3,4.5) | 0.004 |
| Marchant | 54 | 19 | 0.5(0.3,1) | 0.045 | 0.6(0.3,1.2) | 0.14 |
| Govt employee | 35 | 19 | 0.8(0.4,1.5) | 0.5 | 1.1(0.5,2) | 0.96 |
| Private employe | | | | | | |
| Farmer | 142 | 63 | 0.4(0.2,0.5) | 0.00 | 0.5(0.3,1.1) | 0.06 |
| **Insurance** | 97 | 113 | 1 | | 1 | |
| Yes | | | | | | |
| No | 155 | 78 | 2.4(1.6, 3.5) | 0.00 | 5.5(2.1,14) | 0.01 |
| **Transportation availability** | 82 | 98 | 1 | | 1 | |
| Available | | | | | | |
| Not available | 38 | 13 | 1 | | 1 | |
| **Pain** | 199 | 163 | 0.4(0.1,0.3) | 0.00 | 0.4(0.2,1) | 0.01 |
| Yes | | | | | | |
| No | 68 | 73 | 1.8(1.2,2.6) | 0.007 | 1.7(1.1,2.7) | 0.02 |
| **Goal of treatment** | 168 | 103 | 1 | | 1 | |
| Curative | | | | | | |
| Palliative | 154 | 83 | 2.3(1.7,8.5) | 0.04 | 2.1(1.9,11.3) | 0.03 |
| **Social support** | 79 | 100 | 1 | | 1 | |
| Good | | | | | | |
| Poor | | | | | | |

## Discussion

Data on IV chemotherapy adherence is scarce in African countries, including Ethiopia. In this study, the overall magnitude of IV adherence among cancer patients was found to be 176/413(42.6%). Factors identified as independent predictors of chemotherapy adherence include being married, being employed, having transportation availability, and having a curative goal of treatment.

This finding was lower than the adherence rates reported in other studies:, 57% in the USA [12], 90% in Atlanta [10], 90% in the UK [2], 52.6% in France, in Tokyo, Japan, 56.4% [16] and 80.9% in Africa for cytotoxic chemotherapies s [5]. In Uganda 65% adhered to cancer chemotherapy [17], Black Lion Specialized Hospital 83.5% were adherent to their chemotherapy [4]. This study was congruent with a Previous study done in Ethiopia Amhara region 42.3% of patients adhering to chemotherapy [4], and higher than the study done in Catalonia Spain 10% [1].

These studies emphasize the widespread issue of chemotherapy adherence and highlight the urgent need for effective interventions to tackle this critical problem. Varying levels of access to chemotherapy, particularly among individuals with low socio-economic status (SES), play a significant role. Additionally, factors such as poor perception of cancer treatment, limited healthcare access, socio-economic status, educational attainment, and cultural practices all contribute to the differing adherence rates to chemotherapy.

Current findings indicate that being married is associated with a 4.3 times higher likelihood of adhering to IV chemotherapy, similar to previous studies done in North Carolina [18] and the USA, [19], as well as a systemic review [20]. Additionally, government employees were 2.4 times more likely to adhere to chemotherapy, which aligns with the studies from Nigeria [21], the USA [8] and Thailand [22]. This could be due to married patients tend to report more social support, and patients with more social support generally exhibit better medication adherence.

In this study, transportation availability was associated with a 5.5 times higher likelihood of good adherence to chemotherapy, consistent with previous studies conducted in the USA [23], Uganda [24], and North Carola [25]. The unavailability and cost of transportation to the hospital are critical concerns for cancer patients, as they can limit both adherence to cancer therapy and access to care. When resources are scarce, transportation costs compete with other important personal and family needs. Cancer treatment itself incurs high costs, and the additional burden of transportation availability further complicates chemotherapy adherence for cancer patients.

A current study showed that patients who started treatment with a curative goal were 1.7 times more likely to exhibit good adherence to chemotherapy, aligning with previous studies conducted in the Czech Republic [26], and Saudi Arabia [27]. This may be because palliative patients often experience more severe symptoms and a lower quality of life, which can affect their ability to adhere to medication regimens. Additionally, they may have diminished hope for survival, leading to decreased motivation to continue treatment. There might also be reluctance to expose their families to financial burdens for treatments perceived as unlikely to cure the disease. Furthermore, a deep-rooted perception in the community that cancer is incurable may negatively impact patients' adherence levels.

The current finding indicates that, patients with good social support were 2.1 times more likely to adhere to chemotherapy, similar to results from a previous meta-analysis [28]. This may be due to Social support playing a crucial role in medication adherence as it encompasses the assistance individuals receive from their social network, including friends, family, and neighbors, during treatment. It is described as the extent to which individuals have access to assistance from their social network during treatment, which includes friends, family, and

neighbors. Social support can help to minimize psychological stress and increase social adaptability. Social support favors medicine adherence, and therapies that combine social support with psychological interventions can help improve adherence.

Chemotherapy adherence is essential for achieving optimal clinical outcomes, as non-adherence can lead to disease progression, recurrence, and increased mortality rates [3,4]. Clinically, patients who fail to complete their chemotherapy regimens may experience reduced treatment efficacy, higher rates of drug resistance, and poorer overall survival [6]. It has a crucial public health impact, poor adherence contributes to increased healthcare costs due to more frequent hospitalizations and the need for additional interventions [6,10,19].

In low-resource settings like Ethiopia, financial constraints, limited access to treatment centers, and lack of patient awareness are major barriers to adherence [29]. Furthermore, severe side effects, psychological distress, and inadequate social support further contribute to treatment discontinuation [5,10,29]. Addressing these challenges through patient education, financial assistance, and digital adherence tools can improve compliance, ultimately enhancing both individual and population-level cancer outcomes [4].

This study has some limitations. The cross-sectional design of the study creates uncertainty regarding the temporal relationship between exposure and outcome variables. Methodological issues, such as face-to-face interviews, may introduce social desirability bias and recall bias, making it challenging to establish cause-and-effect relationships.

## Conclusions

The study concluded that chemotherapy adherence among cancer patients in Hawassa City was significantly lower compared to published results. Adherence was positively associated with factors such as government employment, curative treatment goals, transportation availability, and marital status. Integrating adherence into existing cancer treatment programs and educating patients about the consequences of non-adherence are essential. Healthcare professionals should assess adherence as part of their evaluations and consider psychological well-being and financial toxicity. Involving family members and enhancing patients' understanding of their condition is crucial. Early diagnosis and initiating curative treatment, along with promoting additional social support from professional agencies and community organizations, can further improve adherence.

## Acknowledgment

We would like to thank data collectors and supervisors for their outstanding contributions. We acknowledge the assistance of Chat GPT in refining the language of this work.

## Consent for publication

Not applicable.

## Author contributions

**Conceptualization:** Bargude Balta, Alemayehu Delelegn, Gulema Demissie, Bedilu Deribe.

**Data curation:** Bargude Balta, Alemayehu Delelegn, Bedilu Deribe.

**Formal analysis:** Bargude Balta, Gulema Demissie.

**Funding acquisition:** Alemayehu Delelegn.

**Investigation:** Alemayehu Delelegn, Gulema Demissie.

**Methodology:** Bargude Balta, Alemayehu Delelegn.

**Project administration:** Alemayehu Delelegn, Bedilu Deribe.

**Resources:** Alemayehu Delelegn.

**Software:** Bargude Balta.

**Supervision:** Gulema Demissie, Bedilu Deribe.

**Validation:** Alemayehu Delelegn, Bedilu Deribe.

**Visualization:** Bargude Balta, Alemayehu Delelegn.

**Writing – review & editing:** Bargude Balta.

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
