## [Decision Letter · Decision Letter 0]

22 Dec 2024

PONE-D-24-52523Adherence to chemotherapy and associated factors among patients with cancer at Hawassa University Comprehensive Specialized Hospital Cancer Treatment Center, Sidama Region, Southern EthiopiaPLOS ONE

Dear Dr. Balta,

Thank you for submitting your manuscript to PLOS ONE. After careful consideration, we feel that it has merit but does not fully meet PLOS ONE’s publication criteria as it currently stands. Therefore, we invite you to submit a revised version of the manuscript that addresses the points raised during the review process.

A rebuttal letter that responds to each point raised by the academic editor and reviewer(s). You should upload this letter as a separate file labeled 'Response to Reviewers'.A marked-up copy of your manuscript that highlights changes made to the original version. You should upload this as a separate file labeled 'Revised Manuscript with Track Changes'.An unmarked version of your revised paper without tracked changes. You should upload this as a separate file labeled 'Manuscript'

We look forward to receiving your revised manuscript.

Kind regards,

Tebelay Dilnessa, MSc

Academic Editor

PLOS ONE

Additional Editor Comments:

The paper generally requires intensive revision. That is, it requires a through edition, revision and proofreading in terms of typographically, punctuation and grammatically.

**Abstract**

The ‘introduction’ should be replaced by ‘Background’.………………… employed from May 20, 2024 to August 31, 2024.government employee, transportation availability, good social supportIt is better use formal and uniform use of AOR, 95%CI throughout the document including the abstract.In the abstract and result, the absolute number (numerator and denominator) is needed together with the percentage. For example, A/B (C%).Your conclusion: - It's important to educate on the impact of non-adherence on survival and disease progression in palliative chemotherapy patients. Have you assessed the knowledge of study participants?Your ‘Data availability’ statement: - All relevant data are within the manuscript and its supporting information files. If so, please attach a supporting file as ‘supplementary file’.

**Introduction**

In the introduction part, the rationale is not explained well. Why you did this research? What has been done before, what is known before and which you intend to fill?

**Method**

A separate ‘Ethical consideration’ and ‘Operational definition’ will be needed. For example, Adherence to chemotherapy, etc.

**Results**

Citation of tables and figures should be standardized.Table and figure descriptions should be described in terms of person, place and time.

**Discussion**

Appropriate paragraphing is needed for this part.Good discussion, but it requires scientific explanation for the variations of results to other studies, not merely comparison.

**Conclusion**

It was beyond your scope and requires revision.

**Others**

The references were not written properly. All references should be written correctly according to Vancouver style.Follow the standard binomial nomenclature, italize journal name and the word ‘et al’Follow the guideline for manuscript writing protocol for PLoS One.

Reviewers' comments:

Reviewer's Responses to Questions

**Comments to the Author**

1. Is the manuscript technically sound, and do the data support the conclusions?

Reviewer #1: Yes

Reviewer #2: Partly

2. Has the statistical analysis been performed appropriately and rigorously? 

Reviewer #1: Yes

Reviewer #2: I Don't Know

3. Have the authors made all data underlying the findings in their manuscript fully available?

Reviewer #1: Yes

Reviewer #2: Yes

4. Is the manuscript presented in an intelligible fashion and written in standard English?

Reviewer #1: No

Reviewer #2: No

5. Review Comments to the Author

Reviewer #1: Thank you for giving me the opportunity to participate as a reviewer.

Generally

your manuscript is good. Has many grammar, clarity, and fluency issues. However, the authors need to improve their work and make it suitable for the scientific community in a short period by positively addressing the reviewers' comments.

Title

The title is very important as it addresses the community, the severity of cancer, and government-related issues.

Abstract

Generally, well structured

Abstract methods:

• You mentioned that your sampling method was purposive. Do you think a non-randomized sampling technique enables you to make inferences?

• You have employed the statistical analysis via multivariate analysis. However, it should be written as multivariable analysis.

Abstract conclusion:

The conclusion needs to be written based on your objectives. Therefore, you should first address the conclusion regarding the magnitude of chemotherapy adherence, followed by the conclusions related to your second objective of identifying factors associated with adherence. Finally, based on your findings, discuss the clinical and public health implications. Accordingly, your conclusion requires further improvement. Additionally, please rewrite the conclusion to ensure it aligns with your findings and compares them with international, regional, and national plans and actual experiences.

Introductions

Your introduction section was written in poor way and need much improvement.

Each paragraph is too long. Please reduce each to 3–4 lines, making the content short and precise.

You did not follow the appropriate coherence for the statement of the problem. The first paragraph should define the problem, the second should address the severity of the problem from a global to a national perspective, the third should discuss the actual and potential burden, the fourth should outline the factors associated with adherence, and the fifth should detail intervention experiences concerning the problem. Finally, the research gap should be clearly identified to convince the scientific community, and the rationale for your study should be well-articulated.

Methods and materials

A study design: voids redundancy words and can be written as institutional based cross-sectional study was employed.

Source of population; Please do not limit the source population to only the Sidama region. Consider including patients from the Guji and Western Arsi zones in the Oromia region and other referral areas in southern Ethiopia. Therefore, ensure it reflects the actual experience.

Sample size and sampling technique: if you have included all patients in cancer center no need of estimating sample size. I need explanation that does Purposive sampling enables us to make inference?

Social support needs more explanation, including the scoring system and the criteria for classifying patients as having good or poor social support.

Result

• Please make the sociodemographic text results concise.

• What are your bases for classifying monthly income?

• Your tables did not comply with the journal's standards. Please, comply with journal standard.

• The social support table needs improvement, such as writing the research question in a shorter, more precise manner, and ensuring it is suitable for the readers.

• Magnitude of adherence: please write and report only the good adherence.

• You did not write statistical interpretation in the factor report.

Discussion

• The introductory statement of the discussion needs to impressive.

• First, you need to discuss your first objective finding.

• write the possible explanation of why your figure is higher or lower or similar with other results

• Your discussion of the second objective (factors) was very poor. First, compare your findings with those of others, then provide possible explanations, and finally, discuss the clinical and public health implications.

Conclusion

• The conclusion statement needs to be written by comparing your results with the plans and outcomes of international, regional, and national experiences. Therefore, avoid the published result.

• You have included too many unnecessary personal opinions. Therefore, the recommendation statement should be based on your findings.

References

Make it compliance with the journal standard and includes DOI number.

Reviewer #2: Thank you for allowing me to review this paper. It addresses an important topic: adherence to cancer treatment.

There are several things that are unclear to me.

Major concerns:

the definition of adherence needs to be spelled out, is it adherence to systemic IV chemotherapy? adherence to oral chemotherapy? both? and why if systemic chemotherapy is that not taken from the chart (as no-shows) but with the Morisky tool as clearly to recruit these patients they had to be in hospital?

Throughout the paper it is not defined and that is crucial for understand adherence to what?

In terms of study data collection, there are a lot of variables described, but only for social support the tool is described how it is measured. Please include how each of these variables were measured and used in the analyses.

Also please clarify the consent process, it is not in the paper, but the ethics statement is confusing, saying written consent was obtained as well as saying consent was not required as it was a retrospective chart review. And that also needs to be clarified in the methods, as clearly data came from the chart.

I would strongly suggest you use a native speaker to review the paper. There are many unclear sentences and the tense should be past tense (the study is completed) and it is written like a protocol, that the work still needs to happen.

There are also a lot of duplicate sentences, such as the patient population, study subjects and inclusion criteria, it is all the same information repeated three times. So please review

In terms of the background, you refer to other studies but no references are provided. So please review and add why not. And I think it could be more succinct .

On page 2 there is a whole section on pts stoppen appropriately, that paragraph needs to be reviewed as it is totally unclear what you mean for me.

In terms of study area, I think some of the details can go, it is irrelevant what lattitude etc the hospital is at.

The sample size calculation section could be more clearly worded, it seems it is based on another study but it is not completely clear to me and for sampling it says all patients were included in the period, but the period is not described and did no one decide to refuse to participate?

And for the analysis, it is not multivariate but multiple variable logistic regression analysis. Can you please describe how you decided which variables in what order you put them in the multiple variable model?

and what were the model fit statistics as there is no AUC/ ROC or anything reported?

And for table 1 why are all aged 51 and over grouped together?

For the monthly income can you report it in US dollar for all readers?

How was missing data handled?

Foor the social support table, I think you can keep percentage and remove the frequency but I am not sure why this data is presented as the variable was categorized in the statistical analyses?

for table 6 there are some cells with <10 persons, those categories need to be collapsed with others,

and why is income not included in these models? in the introduction you describe how income is the most important factor for adherence in similar studies in Africa and it was collected?

I think the discussion and conclusion need to be looked at after the English has been corrected, there are too many sentences unclear to me.

6. PLOS authors have the option to publish the peer review history of their article (what does this mean? ). If published, this will include your full peer review and any attached files.

**Do you want your identity to be public for this peer review?** For information about this choice, including consent withdrawal, please see our Privacy Policy .

Reviewer #1: **Yes: ** Muluken Amare Wudu

Reviewer #2: No

---

## [Author Response · Author response to Decision Letter 0]

2 Feb 2025

January, 5, 2025

Tebelay Dilnessa, MSc

Academic Editor

PLOS ONE

Re: Submission of revised manuscript PONE-D-23-23059

Dear Ms. Tebelay Dilnessa and reviewers,

We greatly appreciate the reviewers for their constructive comments and suggestions. Please find the attached point-by-point response letter to the reviewers’ comments on the manuscript “Adherence to chemotherapy and associated factors among patients with cancer at Hawassa University Comprehensive Specialized Hospital Cancer Treatment Center, Sidama Region, Southern Ethiopia” with PONE-D-24-52523, for consideration. We have carefully reviewed the comments and we have corrected the manuscript accordingly. We hope that the revised version is now suitable for publication, and we look forward to hearing from you in due course.

Sincerely,

Bargude Balta (MSc, MPH)

Corresponding Author

Hawassa University college of health science Cancer Center

Hawassa, Ethiopia

Email: barjuda@gmail.com

WhatsApp: +251919739646

Point-by-Point Responses to Editors' Comments

1. The ‘introduction’ should be replaced by ‘Background’.

Response to 1: Thank you editor for requesting revision. We replaced the introduction with a background

2. ………………… employed from May 20, 2024, to August 31, 2024.

Response to 2: We apologize for our mistake; we paraphrased

3. It is better to use formal and uniform use of AOR, 95%CI throughout the document including the abstract.

Response 3: Thank you, editor, for your comment we edited AOR throughout the document

4. In the abstract and result, the absolute number (numerator and denominator) is needed together with the percentage—for example, A/B (C%).

Response 4: Thank you very much editor we revised based on your comments

5. Your conclusion: - It's important to educate on the impact of non-adherence on survival and disease progression in palliative chemotherapy patients. Have you assessed the knowledge of study participants

Response 5: Thank you, editor, for your constructive comments, we revised the conclusion section of the abstract.

6. Your ‘Data availability’ statement: - All relevant data are within the manuscript and its supporting information files. If so, please attach a supporting file as a ‘supplementary file’.

Response 6: Thank you for asking for revision, this section was revised in both the document and submission system

Introduction

7. In the introduction part, the rationale is not explained well. Why you did do this research? What has been done before, what is known before, and which do you intend to fill?

Response 7: Thank you very much for your requests; we revised the rationale of the study

Point-by-point responses to reviewers’ 1 questions and comments

1. Abstract methods:

You mentioned that your sampling method was purposive. Would a non-randomized sampling technique enable you to make inferences?

Response 1: While a non-randomized sampling technique may limit the generalizability of findings to the broader cancer patient population, it remains a valuable approach for studying adherence to chemotherapy and its associated factors. In this study, a non-randomized sampling method was chosen to ensure the inclusion of patients actively undergoing chemotherapy, allowing for a more focused examination of adherence patterns and influencing factors. Given the nature of adherence studies, randomization may not always be feasible, as patient selection often depends on treatment accessibility, willingness to participate, and healthcare facility settings. We applied rigorous inclusion and exclusion criteria to minimize potential biases, ensured sample diversity, and used statistical adjustments to control for confounding variables. While our findings provide meaningful insights into adherence behavior, we acknowledge the need for cautious interpretation and recommend further research with randomized approaches to enhance external validity.

2. You have employed statistical analysis via multivariate analysis. However, it should be written as a multivariable analysis.

Response 2: thank you. We corrected it.

3. Abstract conclusion: The conclusion needs to be written based on your objectives. Therefore, you should first address the conclusion regarding the magnitude of chemotherapy adherence, followed by the findings related to your second objective of identifying factors associated with adherence. Finally, based on your findings, discuss the clinical and public health implications. Accordingly, your conclusion requires further improvement. Additionally, please rewrite the conclusion to ensure it aligns with your findings and compares them with international, regional, and national plans and actual experiences.

Response 3: Thank you, reviewer, we revised the entire conclusion

4. Your introduction section was written poorly and needs much improvement.

Each paragraph is too long. Please reduce each to 3–4 lines, making the content short and precise. You did not follow the appropriate coherence for the statement of the problem. The first paragraph should define the problem, the second should address the severity of the problem from a global to a national perspective, the third should discuss the actual and potential burden, the fourth should outline the factors associated with adherence, and the fifth should detail intervention experiences concerning the problem. Finally, the research gap should be identified to convince the scientific community, and the rationale for your study should be well-articulated.

Response 4: Thank you very much reviewer for your constructive comments: we revised the entire introduction

5. A study design: that voids redundancy words and can be written as an institutional-based cross-sectional study was employed.

Response 5: we appreciate this feedback and have revised

6. Source of population; Please do not limit the source population to only the Sidama region. Consider including patients from the Guji and Western Arsi zones in the Oromia region and other referral areas in southern Ethiopia. Therefore, ensure it reflects the actual experience.

Response 6: Thank you, reviewer, I appreciate your effort, I fully accepted and revised the source population

7. Sample size and sampling technique: if you have included all patients in the cancer center no need to estimate the sample size. I need an explanation does Purposive sampling enable us to make inferences?

Response 7: Purposive sampling, while not a probability-based method, can still enable meaningful inferences, particularly in studies focusing on specific populations, such as cancer patients undergoing chemotherapy. In our study, purposive sampling was employed to ensure the inclusion of eligible participants actively receiving treatment at the cancer center, allowing us to examine adherence patterns and associated factors within this defined group. This approach enhances the study's internal validity by capturing relevant data from those directly experiencing the phenomenon under investigation. However, we acknowledge that purposive sampling may limit external validity and generalizability to broader cancer populations. To mitigate this, we ensured sample diversity and applied statistical techniques to address potential biases. While probability sampling would strengthen generalizability, purposive sampling remains an appropriate choice for in-depth analysis of adherence behaviors in a targeted clinical setting."

8. Social support needs more explanation, including the scoring system and the criteria for classifying patients as having good or poor social support.

Response 8: Thank you, reviewer, for your request for social support. we used the MSPSS tool, which has 12 questions. We used to mean of 12 questions those who responded above the mean indicated greater perceived social support and those below the mean coded as poor social support. And indicated under the operational definition.

Result

9. Please make the sociodemographic text results concise.

Response 9: Thank you very much for the comments we edited sociodemographic texts

10. What are your bases for classifying monthly income?

Response 9; Thank you, reviewer, for your request on the monthly income category, we categorized it based similar study baseline. We used the category of World Bank income categorization of 2024. Hence (category of WB annual income: low income, <=$995; lower middle income, $996–3,945; upper middle income, $3,946–12,195; and high income, $12,196 or more.). Based on our data the category is low-income and lower-middle-income, hence in our case there is no upper-middle-income and high-income category.

11. Your tables did not comply with the journal's standards. Please, comply with journal standards.

Response 11: Thank you editor your genuine comments, we edited tables respect with to journal guideline

12. The social support table needs improvement, such as writing the research question in a shorter, more precise manner, and ensuring it is suitable for the readers.

Response 12: Thank you, reviewers, The questions under the tool MSPSS we used for social support were somewhat longer; by considering the originality of the tool we minimized some words suitable for readers.

13. Magnitude of adherence: please write and report only the good adherence.

Response 13; Thank you very much reviewer for your comment we revised the magnitude of the adherence report

Discussion

14.. The introductory statement of the discussion needs to be impressive. First, you need to discuss your first objective finding. write the possible explanation of why your figure is higher or lower or similar with other results.

Your discussion of the second objective (factors) was very poor. First, compare your findings with those of others, then provide possible explanations, and finally, discuss the clinical and public health implications.

Response 14: Response: we appreciate this feedback and have reviewed the entire discussion and edited it.

14. Make it compliant with the journal standard and include DOI number.

Response 14: Thank you editor for your request to edit the journal standard and DOI. We involved DOI for all references except books and reports.

Point-by-point responses to Reviewer #2 questions and comments

1. The definition of adherence needs to be spelled out, is it adherence to systemic IV chemotherapy? adherence to oral chemotherapy? both? and why if systemic chemotherapy was not taken from the chart (as no-shows) but with the Morisky tool as clearly to recruit these patients they had to be in the hospital?

Response 1: Thank you, reviewer, for your detailed comments. We revised the definition of adherence to Adherence to IV chemotherapy: and took patient data from those admitted to inpatient care for intravenous chemotherapy. That means from both the patients and patient chart. Under the data collection procedure section, we included data collected from patient charts and interviews.

2. Throughout the paper it is not defined and that is crucial for understanding adherence to what?

Response 2: Thank you very much we operationalized adherence to IV chemotherapy.

3. In terms of study data collection, there are a lot of variables described, but only for social support the tool is described how it is measured. Please include how each of these variables were measured and used in the analyses.

Response 3: "We appreciate the reviewer's insightful comment. We edited the entire data collection section. A detailed description of measurement tools and variable categorization has been included in the revised manuscript for clarity.

4. Also please clarify the consent process, it is not in the paper, but the ethics statement is confusing, saying written consent was obtained as well as saying consent was not required as it was a retrospective chart review. And that also needs to be clarified in the methods, as clearly data came from the chart.

Response 4: Thank you, editor, for your comments we apologize for our mistakes, we interviewed patients so we took consent from the patient and removed this part from ethics “Consent was not required as it was a retrospective chart review”

5. I would strongly suggest you use a native speaker to review the paper. There are many unclear sentences and the tense should be past tense (the study is completed) and it is written like a protocol, that the work still needs to happen. There are also a lot of duplicate sentences, such as the patient population, study subjects, and inclusion criteria, it is all the same information repeated three times. So please review

Response 5: Thank you, reviewer, we revised the entire manuscript for language and any related editing

6. In terms of the background, you refer to other studies but no references are provided. So please review and add why not. And I think it could be more succinct.

Response 6: Thank you very much, we revised the entire background.

7. On page 2 there is a whole section on pts stopped appropriately, that paragraph needs to be reviewed as it is unclear what you mean for me. In terms of the study area, I think some of the details can go, it is irrelevant what latitude, etc the hospital is at.

Response 7: Thanks, reviewer, for your comments, we reviewed the methodology and removed unnecessary descriptions.

8. The sample size calculation section could be more clearly worded, it seems it is based on another study but it is not completely clear to me for sampling it says all patients were included in the period, but the period is not described, and did no one decide to refuse to participate?

Response 8: Thank you very much reviewer we used the proportion of chemotherapy adherence done in a previous study in the Amhara region Ethiopia, to compute the sample size calculation and we cited it. Regarding the study period, we mentioned it as a study area and the period section. Our study period was “May 20 to August 31, 2024”. All of our respondents responded survey

9. And for the analysis, it is not multivariate but multiple variable logistic regression analysis. Can you please describe how you decided which variables in what order you put them in the multiple variable model? What were the model fit statistics as there is no AUC/ ROC or anything reported?

Response 9: Selection of Variables for the Multiple Variable Logistic Regression Model:

Variables were selected for inclusion in the multiple variable logistic regression model based on their clinical relevance, statistical significance in the bivariate analysis (p-value ≤ 0.25), and prior evidence from existing literature. Initially, a bivariate logistic regression was conducted to identify candidate variables associated with the case magnitude. Variables with a significant association at the bivariate level, along with key demographic and risk factors known to influence adherence, were included in the multiple variable models.

Order of Variable Inclusion:

A hierarchical approach was used, beginning with demographic factors (such as marital status, and occupation), followed by other factors such as insurance and Transportation availability This stepwise approach ensured that potential confounders were controlled while assessing the independent effects of each variable.

Model Fit Statistics:

Although AUC/ROC analysis was not explicitly reported, model fit was assessed using statistical measures such as the Hosmer-Lemeshow goodness-of-fit test to evaluate calibration, and the likelihood ratio test to compare nested models. Additionally, the Nagelkerke R² was used to estimate the proportion of variance explained by the model. If required, further model discrimination metrics such as AUC/ROC can be computed to strengthen the validity of the findings.

10. And for table 1 why are all aged 51 and over grouped?

For the monthly income can you report it in US dollars for all readers?

Response 10; Thank you reviewer we have very few patients above 60 so we merged them and categorized ages above 50. We changed the monthly income to US dollars for all reviewers.

11. How was missing data handled?

Response 11: Missing data was handled using appropriate statistical techniques to minimize potential biases. Cases with incomplete or missing key variables were carefully examined, and where feasible, missing

---

## [Decision Letter · Decision Letter 1]

18 Feb 2025

PONE-D-24-52523R1Adherence to chemotherapy and associated factors among patients with cancer at Hawassa University Comprehensive Specialized Hospital Cancer Treatment Center, Sidama Region, Southern EthiopiaPLOS ONE

Dear Dr. Balta,

Thank you for submitting your manuscript to PLOS ONE. After careful consideration, we feel that it has merit but does not fully meet PLOS ONE’s publication criteria as it currently stands. Therefore, we invite you to submit a revised version of the manuscript that addresses the points raised during the review process.

We look forward to receiving your revised manuscript.

Kind regards,

Tebelay Dilnessa, MSc

Academic Editor

PLOS ONE

Journal Requirements:

Additional Editor Comments:

- The figure (s) should be prepared without description in TIF version and uploded as separate file. The description should be put both within the main manuscript where the figure cited and below the list of references.

- The document should be proofread and keep its scientific quality.

- The authors contribution should be removed from the main manuscript because the system already will create it.

Reviewers' comments:

Reviewer's Responses to Questions

**Comments to the Author**

1. If the authors have adequately addressed your comments raised in a previous round of review and you feel that this manuscript is now acceptable for publication, you may indicate that here to bypass the “Comments to the Author” section, enter your conflict of interest statement in the “Confidential to Editor” section, and submit your "Accept" recommendation.

Reviewer #1: All comments have been addressed

Reviewer #2: (No Response)

2. Is the manuscript technically sound, and do the data support the conclusions?

Reviewer #1: Yes

Reviewer #2: No

3. Has the statistical analysis been performed appropriately and rigorously? 

Reviewer #1: Yes

Reviewer #2: No

4. Have the authors made all data underlying the findings in their manuscript fully available?

Reviewer #1: Yes

Reviewer #2: No

5. Is the manuscript presented in an intelligible fashion and written in standard English?

Reviewer #1: Yes

Reviewer #2: No

6. Review Comments to the Author

Reviewer #1: The manuscript has improved following the revision.

Title

Need to include IV

Introduction

The introduction requires improvement, including the addition of the following elements: the magnitude of chemotherapy adherence, consolidation of burden statements into a single paragraph, discussion of factors associated with chemotherapy adherence, addressing interventions, and a clearer articulation of the research gap.

Discussion

The discussion need improvement particularly, clinical and public health implications

The possible explanation or speculation need citation

Reviewer #2: Thank you for the revision. I would have preferred to see a tracked changes version. A lot of comments have not be appropriately addressed.

For example, the introduction has not changed much but the focus now makes less sense. In the introduction the authors bring in the concept of relative dose received. This has everything to do with the tolerability of treatment. A patient may receive dose reduced treatment becuase of side effects and could have totally not be related to patients not showing up for treatment (AKA non-adherence. In addition, they still use the MOrisky to measure adherence which still is focused on adherence to medication in pill bottles, not chemo given as IV and not dose received.

7. PLOS authors have the option to publish the peer review history of their article (what does this mean? ). If published, this will include your full peer review and any attached files.

**Do you want your identity to be public for this peer review?** For information about this choice, including consent withdrawal, please see our Privacy Policy .

Reviewer #1: **Yes: ** Muluken Amare wudu

Reviewer #2: No

---

## [Author Response · Author response to Decision Letter 1]

3 Mar 2025

March, 1, 2025

Tebelay Dilnessa, MSc

Academic Editor

PLOS ONE

Re: Submission of revised manuscript PONE-D-23-23059

Dear Ms. Tebelay Dilnessa and reviewers,

We greatly appreciate the reviewers for their constructive comments and suggestions. Please find the attached point-by-point response letter to the reviewers’ comments on the manuscript “Adherence to chemotherapy and associated factors among patients with cancer at Hawassa University Comprehensive Specialized Hospital Cancer Treatment Center, Sidama Region, Southern Ethiopia” with PONE-D-24-52523, for consideration. We have carefully reviewed the comments and we have corrected the manuscript accordingly. We hope that the revised version is now suitable for publication, and we look forward to hearing from you in due course.

Sincerely,

Bargude Balta (MSc, MPH)

Corresponding Author

Hawassa University college of health science Cancer Center

Hawassa, Ethiopia

Email: barjuda@gmail.com

WhatsApp: +251919739646

Point-by-Point Responses to Editors' Comments

1. The figure (s) should be prepared without description in the TIF version and uploaded as a separate file. The description should be put both within the main manuscript where the figure is cited and below the list of references.

Response to 1: Thank you editor for requesting revision. We uploaded it without description and placed it in the main manuscript where the figure is cited and after reference

2 The document should be proofread and keep its scientific quality.

Response to 2: We proofread the document while ensuring that its scientific quality and integrity are maintained."

3 The author's contribution should be removed from the main manuscript because the system already created it.

Response 3: Noted. The authors' contribution section will be removed from the main manuscript, as the system will generate it automatically."

Point-by-point responses to Reviewer #1 questions and comments

His comments:

1. The introduction requires improvement, including the addition of the following elements: the

magnitude of chemotherapy adherence, consolidation of burden statements into a single

paragraph, discussion of factors associated with chemotherapy adherence, addressing

interventions, and a clearer articulation of the research gap.

Response 1: Thank you, reviewer, for revisiting and commenting on our manuscript we revised and addressed the issue in the 2nd and 3rd paragraphs.

2. The discussion needs improvement, particularly regarding clinical and public health implications. Possible explanations or speculations need to be cited.

Response 2: Thank you for your feedback. I agree that the discussion section could benefit from further elaboration on the clinical and public health implications. It is essential to contextualize the findings by exploring the potential impact on healthcare delivery and patient outcomes. We added two paragraphs under the discussion section before the limitation of study which can address the issue.

Reviewer #2: Thank you for the revision. I would have preferred to see a tracked changes version. A lot of comments have not been appropriately addressed.

Response: Thank you for your feedback. I acknowledge that some comments may not have been fully addressed, and I sincerely appreciate your patience. I will carefully review all the provided comments to ensure they are appropriately incorporated. If there are specific areas that require further clarification or revision, please let me know, and I will make the necessary improvements to strengthen the work. Your insights are valuable in enhancing the quality and rigor of the discussion. Here we revisited our revision reviewer comments there were highlights of revision in the main manuscript.

1. The definition of adherence needs to be spelled out, is it adherence to systemic IV chemotherapy? adherence to oral chemotherapy? both? and why if systemic chemotherapy was not taken from the chart (as no-shows) but with the Morisky tool as clearly to recruit these patients they had to be in the hospital?

Response 1: Thank you, reviewer, for your detailed comments. We acknowledge reviewer comments that The Morisky Medication Adherence Scale (MMAS) was originally designed to assess adherence to oral medications, particularly for chronic diseases like hypertension and diabetes. While it has been widely used in various medical contexts, its applicability to intravenous (IV) chemotherapy adherence. We put previous studies done using the Morisky Medication Adherence Scale to measure adherence to IV chemotherapy. “Bekalu YE, Wudu MA, Gashu AW. Adherence to Chemotherapy and Associated Factors Among Patients With Cancer in Amhara Region, Northeastern Ethiopia, 2022. A Cross-Sectional Study. Cancer Control. 2023;30:1–10.”

2. Throughout the paper it is not defined and that is crucial for understanding adherence to what?

Response 2: Thank you very much we operationalized adherence to IV chemotherapy (It has been highlighted in a manuscript operational definition section.

3. In terms of study data collection, there are a lot of variables described, but only for social support the tool is described how it is measured. Please include how each of these variables were measured and used in the analyses.

Response 3: "We appreciate the reviewer's insightful comment. We edited the entire data collection section. A detailed description of measurement tools and variable categorization has been included in the revised manuscript for clarity. We highlighted under the data collection tool section

4. Also please clarify the consent process, it is not in the paper, but the ethics statement is confusing, saying written consent was obtained as well as saying consent was not required as it was a retrospective chart review. And that also needs to be clarified in the methods, as clearly data came from the chart.

Response 4: Thank you, reviewer, for your comments we apologize for our mistakes, we interviewed patients so we took consent from the patient and removed this part from ethics “Consent was not required as it was a retrospective chart review” and revised the remaining section

5. I would strongly suggest you use a native speaker to review the paper. There are many unclear sentences and the tense should be past tense (the study is completed) and it is written like a protocol, that the work still needs to happen. There are also a lot of duplicate sentences, such as the patient population, study subjects, and inclusion criteria, it is all the same information repeated three times. So please review

Response 5: Thank you, reviewer, we revised the entire manuscript for language and any related editing

6. In terms of the background, you refer to other studies but no references are provided. So please review and add why not. And I think it could be more succinct.

Response 6: Thank you very much, we revised the entire background.

7. On page 2 there is a whole section on pts stopped appropriately, that paragraph needs to be reviewed as it is unclear what you mean for me. In terms of the study area, I think some of the details can go, it is irrelevant what latitude, etc the hospital is at.

Response 7: Thanks, reviewer, for your comments, we reviewed the methodology and removed unnecessary descriptions.

8. The sample size calculation section could be more clearly worded, it seems it is based on another study but it is not completely clear to me for sampling it says all patients were included in the period, but the period is not described, and did no one decide to refuse to participate?

Response 8: Thank you very much reviewer we used the proportion of chemotherapy adherence done in a previous study in the Amhara region Ethiopia, to compute the sample size calculation and we cited it. Regarding the study period, we mentioned it as a study area and the period section. Our study period was “May 20 to August 31, 2024”. All of our respondents responded survey

9. And for the analysis, it is not multivariate but multiple variable logistic regression analysis. Can you please describe how you decided which variables in what order you put them in the multiple variable model? What were the model fit statistics as there is no AUC/ ROC or anything reported?

Response 9: Selection of Variables for the Multiple Variable Logistic Regression Model:

Variables were selected for inclusion in the multiple variable logistic regression model based on their clinical relevance, statistical significance in the bivariate analysis (p-value ≤ 0.25), and prior evidence from existing literature. Initially, a bivariate logistic regression was conducted to identify candidate variables associated with the case magnitude. Variables with a significant association at the bivariate level, along with key demographic and risk factors known to influence adherence, were included in the multiple variable models.

Order of Variable Inclusion:

A hierarchical approach was used, beginning with demographic factors (such as marital status, and occupation), followed by other factors such as insurance and Transportation availability This stepwise approach ensured that potential confounders were controlled while assessing the independent effects of each variable.

Model Fit Statistics:

Although AUC/ROC analysis was not explicitly reported, model fit was assessed using statistical measures such as the Hosmer-Lemeshow goodness-of-fit test to evaluate calibration, and the likelihood ratio test to compare nested models. Additionally, the Nagelkerke R² was used to estimate the proportion of variance explained by the model. If required, further model discrimination metrics such as AUC/ROC can be computed to strengthen the validity of the findings.

10. And for table 1 why are all aged 51 and over grouped?

For the monthly income can you report it in US dollars for all readers?

Response 10; Thank you reviewer we have very few patients above 60 so we merged them and categorized ages above 50. We changed the monthly income to US dollars for all reviewers.

11. How was missing data handled?

Response 11: Missing data was handled using appropriate statistical techniques to minimize potential biases. Cases with incomplete or missing key variables were carefully examined, and where feasible, missing values were addressed through multiple imputation or mean substitution methods. Since our missing data were minimal a complete case analysis was performed to ensure the integrity of the findings. Additionally, sensitivity analyses were conducted to assess the impact of missing data on the study results. Efforts were also made during data collection to minimize missing responses by providing thorough training to data collectors and conducting real-time data checks."

12. For the social support table, I think you can keep the percentage and remove the frequency but I am not sure why this data is presented as the variable was categorized in the statistical analyses.

Response 12: Thank you, reviewer, we used the MSPSS tool to measure social support, it has 12 questions/items and reported results. Then we did mean and categorized it as good and poor social support based on the mean. above the mean good social support and below the mean poor social support. Also, we removed frequency.

13. For Table 6 there are some cells with <10 persons, those categories need to be collapsed with others, why is income not included in these models? in the introduction, you describe how income is the most important factor for adherence in similar studies in Africa and how it was collected.

Response 13: Thank you, reviewer, we acknowledge the comments, for each cell more than 5 observations were collapsed in the revised version. Regarding monthly income, we checked it in the bivariable regression cut point p-value of 0.25 but it is not a candidate variable for multivariable regression so we didn’t include it in the factor table even though it is an important factor in Africa.

14. I think the discussion and conclusion need to be looked at after the English has been corrected, there are too many sentences unclear to me.

Response 14: Thank you, reviewer, I appreciate your effort for scientific advances we edited the entire manuscript.

---

## [Decision Letter · Decision Letter 2]

5 Mar 2025

Adherence to intravenous chemotherapy and associated factors among patients with cancer at Hawassa University Comprehensive Specialized Hospital Cancer Treatment Center, Sidama Region, Southern Ethiopia

PONE-D-24-52523R2

Dear Dr. Balta,

We’re pleased to inform you that your manuscript has been judged scientifically suitable for publication and will be formally accepted for publication once it meets all outstanding technical requirements.

Kind regards,

Tebelay Dilnessa, MSc

Academic Editor

PLOS ONE

Additional Editor Comments (optional):

Reviewers' comments:

Reviewer's Responses to Questions

**Comments to the Author**

1. If the authors have adequately addressed your comments raised in a previous round of review and you feel that this manuscript is now acceptable for publication, you may indicate that here to bypass the “Comments to the Author” section, enter your conflict of interest statement in the “Confidential to Editor” section, and submit your "Accept" recommendation.

Reviewer #1: All comments have been addressed

2. Is the manuscript technically sound, and do the data support the conclusions?

Reviewer #1: No

3. Has the statistical analysis been performed appropriately and rigorously? 

Reviewer #1: Yes

4. Have the authors made all data underlying the findings in their manuscript fully available?

Reviewer #1: Yes

5. Is the manuscript presented in an intelligible fashion and written in standard English?

Reviewer #1: Yes

6. Review Comments to the Author

Reviewer #1: (No Response)

7. PLOS authors have the option to publish the peer review history of their article (what does this mean? ). If published, this will include your full peer review and any attached files.

**Do you want your identity to be public for this peer review?** For information about this choice, including consent withdrawal, please see our Privacy Policy .

Reviewer #1: No

---

## [Editor Report · Acceptance letter]

PONE-D-24-52523R2

PLOS ONE

Dear Dr. Balta,

I'm pleased to inform you that your manuscript has been deemed suitable for publication in PLOS ONE. Congratulations! Your manuscript is now being handed over to our production team.

Kind regards,

on behalf of

Dr. Tebelay Dilnessa

Academic Editor

PLOS ONE